# Temporal trends in evidence supporting novel drug target discovery

**Maria J. Falaguera** [1,2] ✉**, Ellen M. McDonagh** [1,2,3]**, David Ochoa** [1,2]**, Polina V. Rusina**[1,2]**, Juan Maria Roldan-Romero**[1,2]**, David G. Hulcoop** [1,2,3]**, Andrew R. Leach** [2] **& Ian Dunham** [1,2,3]

Over the past decade, about one-fifth of FDA-approved drugs each year involve novel mechanism-of-action human targets. Although riskier than modulating well-known targets, these therapies address unmet needs and strengthen sector innovation. The Open Targets Platform is a valuable open resource for identifying novel targets, integrating diverse datasets with regular updates and a user-friendly interface. To expand its capabilities, we implement comprehensive timestamping across millions of biomedical data points and introduce a target novelty metric for disease contexts, enabling discovery of novel targets within the ecosystem. We also present a retrospective analysis of novel drug target approvals over two decades, revealing a shift around 2015: supportive biomedical evidence (e.g., human genetics, literature-derived insights, differential expression, and pathway data) increasingly appears before rather than after the approval year. These findings underscore the importance of time-based evidence assessments for earlier identification of novel clinical opportunities and offer guidance for future target selection trends.

Drug approvals involving novel mechanism-of-action (MoA) human targets (the first time a therapeutic has been approved against that target ever)[1–7] currently account for approximately one-fifth of the new drugs approved annually by the FDA over the past decade[1–12]. Despite the additional risks their development may entail compared to other therapeutics that modulate well-known targets[13–16], the fact that most of these novel MoA drugs are developed to treat diseases where there is an unmet medical need (particularly in oncology and rare diseases[1–7,17,18]) highlights their potential to significantly impact patients' lives and help biopharma companies consolidate a strong innovative position in the sector. Timely identification of these novel drug target opportunities, as evidence of their therapeutic value emerges, is critical for this.

Comprehensive tracking of new biomedical evidence relating a gene or protein to a disease or phenotype is a significant challenge, due to the vast and rapidly expanding volume of potentially relevant data from multiple sources that is being generated and made publicly available since the advent of human genome sequencing[19]. In recent

years, research[20–22], commercial platforms[23,24], and publicly available resources[25–28] have been developed to try and capture biological innovation and attention trends in the pharmaceutical sector by harnessing data from various sources, including scientific publications, pharmaceutical patent claims, clinical trials, and/or research grant applications. In the arena of freely available resources with potential for the identification of novel therapeutic targets, the Open Targets Platform (https://platform.opentargets.org/) occupies a strategic position due to the breadth of data sources that it integrates (e.g. scholarly literature, patent claims, genetic data, animal model experiments and clinical trials), its regular quarterly updates, its systematic, score-based assessment of evidence relevance between targets and diseases, its intuitive web user interface, and its open-source infrastructure that allows use with custom data[29].

With the aim of harnessing and expanding the capabilities of the Open Targets Platform in this realm, we have developed a method that: (i) systematically timestamps available evidence on the Platform, linking potential causal biological targets to diseases; (ii) tracks how

[1]Open Targets, Wellcome Genome Campus, Hinxton, UK. [2]European Molecular Biology Laboratory, European Bioinformatics Institute (EMBL-EBI), Wellcome Genome Campus, Hinxton, UK. [3]Wellcome Sanger Institute, Wellcome Genome Campus, Hinxton, UK. ✉e-mail: mariaf@ebi.ac.uk

the degree of confidence in a target−disease association evolves over time based on the available collective evidence; and (iii) enables the timely identification of novel targets with emerging therapeutic potential in a disease-specific context. Using the results of this approach, we have conducted a retrospective analysis of novel MoA drug approvals over the past two decades, evaluating the breadth, type and timing of the biomedical evidence supporting the underlying target−indication hypotheses. Our analysis reveals a notable shift in trends around 2015, after which time supportive biomedical evidence (e.g., human genetic, literature-derived, differential expression and pathway-related data) appears before rather than after the drug approval year. We believe these findings underscore the value of time-based assessments of biomedical evidence, such as the approach introduced here, in facilitating the earlier identification of innovative drug target opportunities.

## Results

### Timestamping evidence supporting target−disease associations

Our first step was to comprehensively timestamp the 28 million pieces of evidence that comprise the Open Targets Platform ecosystem. These pieces of evidence represent information that supports an association between a human target and a disease indication (Fig. 1a). To determine the date on which the evidence was originally reported or deposited, we have investigated the more than 20 sources of evidence included in the Open Targets Platform. Overall, two categories of timestamps were identified: (1) primary source date; the date of publication in the primary source from which the evidence is mined (e.g. original scientific publication, patent claim, Genome-Wide Association Study (GWAS) or clinical study), and (2) curation date; the date of deposition of the evidence into a repository by an expert curator (e.g. Gene2Phenotype[30], Orphanet[31] or Genomics England (GEL) PanelApp[32]). In total, 99% (27,819,439) of the association evidence

integrated by Open Targets has been dated, including 21 million association evidences from literature sources (i.e., Europe PMC)[33], 4.2 million evidences of association from repositories of genetic association experiments (e.g., GWAS associations), 0.5 million association evidences from sources of approved drugs and clinical candidates (i.e., ChEMBL)[34], and 2 million association evidences from other sources (see the Supplementary Information for a full list). The range of timestamps correspond to the nature of the evidence; for example, those derived from animal model experiments[35] and clinical trials span several decades while those resulting from specific research projects (e.g., Project Score[36] and CRISPR Screen[37]) match the generation and lifetime of the project (Fig. 1b). Overall, most evidence has accumulated after the year 2000, which aligns to the Platform's focus on genetic data sources.

### Temporal profiles for target−disease associations

With the evidence timestamped, we can retrospectively reconstruct the temporal profile of the 3.6 million target−disease associations with supporting evidence in the Open Targets Platform. The assessment presents a quantitative and qualitative analysis of the evolution of supporting biomedical data based on the Open Targets Platform association scoring framework[38]. The scoring framework assigns each target−disease pair a set of harmonised and normalised scores between 0 and 1 that summarise the strength and repetition of evidence supporting the target−disease connection and the level of confidence in its translational value (see the "Methods" section for further details). While the association scores provided in the Platform are calculated based on all evidence currently available for the target −disease pair based upon the latest data release, our temporal assessment involves a recalculation of these scores for each association and each year, considering only evidence accumulated up to that point in time. In Fig. 2, 'Evidence' and 'Association' graphs, we

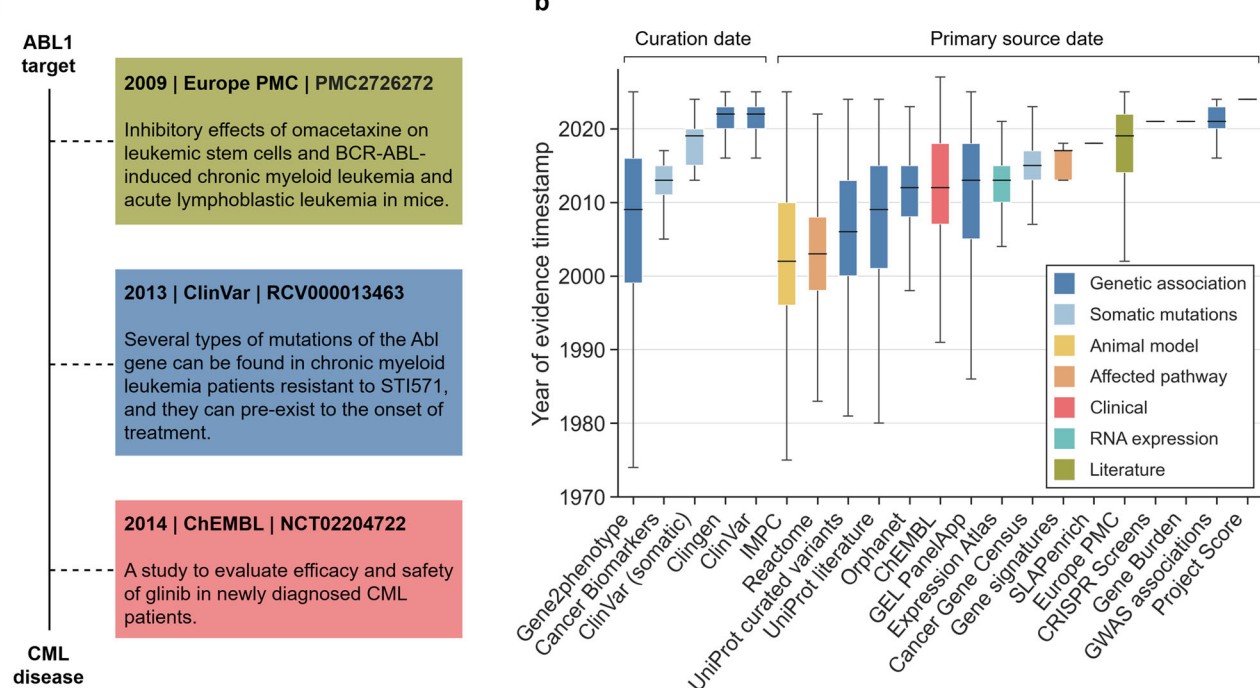

**Fig. 1 | Timestamps of evidence supporting target−disease associations. a** An example of timestamped evidence from three sources supporting a target−disease association in the Open Targets Platform. CML, chronic myelogenous leukaemia. *ABL1*, ABL proto-oncogene 1, non-receptor tyrosine kinase. **b** Distribution of 27,819,439 Platform evidence annotated with timestamps (y-axis), source of origin (x-axis), source category (colour) and timestamp nature (top brackets). IMPC,

International Mouse Phenotyping Consortium. GEL PanelApp, Genomics England PanelApp. GWAS, Genome-Wide Association Studies. See Supplementary Information for a breakdown of the evidence count by data source. Box plots show the median (centre line), the 25th–75th percentiles (box), whiskers extending to the most extreme points within 1.5 × IQR.

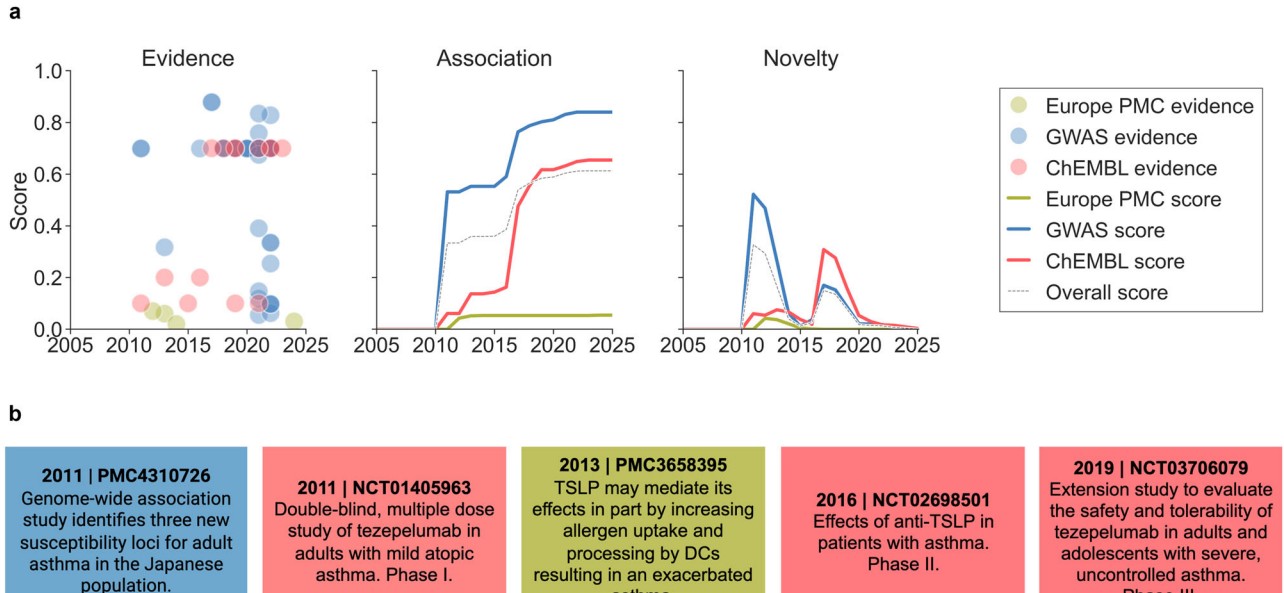

**Fig. 2 | Temporal profiles for the association between thymic stromal lymphopoietin (TSLP) and asthma. a** The 'Evidence' graph shows pieces of evidence supporting the association, mapped to their Open Targets Platform evidence score (y-axis), timestamp (x-axis) and source (colour). The 'Association' and 'Novelty' graphs show how the Platform's source and overall association and novelty scores have evolved over time. **b** Examples of evidence that have triggered shifts in the Platform's association scores and novelty peaks. The identifiers of the reference clinical trials (NCT) and PubMed Central (PMC) publications are shown.

exemplify this for the association between thymic stromal lymphopoietin (TSLP) and asthma, which is supported by literature data ingested from Europe PMC (green), genetic data derived from GWAS (blue) and clinical data provided by ChEMBL (red). For further clarity, Fig. 2b provides examples of evidence. In 2011, Hirota et al.[39] report a GWAS linking TSLP with asthma in adults. This GWAS association is assigned an initial evidence score of 0.70 in the Platform, then following harmonisation and normalisation, a GWAS association source score of 0.53. Also in 2011, a phase I clinical trial (NCT01405963) was initiated in adults with mild atopic asthma to investigate tezepelumab, a human monoclonal antibody with TSLP blocking properties. This is captured as ChEMBL evidence with a score of 0.10 in the Platform. Between 2012 and 2014, three Europe PMC publications suggest the involvement of TSLP in asthma[40–42] and are assigned respective evidence scores of 0.07, 0.06 and 0.02 in the Platform. This is followed by the initiation of Phase II (NCT02698501) and Phase III (NCT03706079) clinical trials in 2016 and 2019, respectively, to further evaluate the efficacy and safety of tezepelumab in treating asthma in adult patients. These trials are recorded in the Platform as ChEMBL evidence, with respective scores of 0.20 and 0.70. Combined with previous clinical evidence, this results in a harmonised and normalised ChEMBL source score of 0.61 in 2019. Aggregating, harmonising and normalising the three source association scores produces an overall association score curve showing two main shifts: one in 2011 corresponding to robust genetic support for the association emerging, and a second in 2017 corresponding to the initiation of advanced clinical trials providing further support for the association.

**Novelty assessment of target−disease associations**
The shifts in association scores described in the previous section reflect instances when new supporting evidence of the target being a potential causal factor of the disease is generated. To quantify this change, we introduce a new 'novelty' metric (see the 'Novelty' graph in Fig. 2a). In essence, the mathematical formula captures shifts in the association score value as peaks of novelty, which subsequently decay until reaching zero as time passes (see the "Methods" section for further details). By relying on the evolution of the association score rather

than the appearance of the earliest piece of evidence as the criterion for claiming novel association, this approach helps prioritise stronger signals of novelty from the background of evidence. Pieces of evidence with low confidence (low score) are assessed more cautiously, whereas more confident signals (high-scoring evidence) are emphasised, even if they appear later. This is exemplified by the low-scoring Europe PMC's pieces of evidence for the TSLP and asthma association between 2012 and 2014, and the corresponding novelty peaks. There may be higher peaks in the future if more relevant publications appear. The reverse scenario is also adequately addressed by this metric, where the initial evidence has a high score and triggers robust peaks, followed by subsequent evidence with a lower, comparable, or higher score. Examples include the GWAS association peaks in 2011 (0.52) and in 2017 (0.17). Furthermore, the ChEMBL timeline exemplifies a combination of the previous two scenarios, depicting multiple peaks corresponding to different clinical phases. We find it convenient to report the different peaks as moments of novelty, as each captures a distinct type of knowledge novelty which is ultimately weighted and contextualised by the novelty score value (see the "Methods" section for more details). In summary, Fig. 2 shows the differences between the accumulation of evidence for a target−disease association and the evolution of the Open Targets Platform association and novelty scores, with novelty peaks providing a clear view of the onset, quality, and quantity of evidence over time.

**Biomedical associations with novelty signals in 2025**
Through our analysis, we have found that 68,012 (2%) out of the 2,914,983 target−disease associations that constitute the Open Targets Platform have novelty peaks in 2025. These associations involve 13,289 (44%) out of the 30,087 unique targets in the Platform, including 12,680 protein-coding genes. The majority of these targets have not yet been explored clinically (11,890; 89%), and 2130 (16%) have a reported binding ligand in ChEMBL. In addition, only 6% (856) of these targets have adverse events annotated in the Platform. Regarding the top therapeutic areas in which these target−disease associations with peaks of novelty in 2025 are found, 41% of them involve oncological diseases (27,577), 9% involve neuronal diseases (6264), and 7% involve

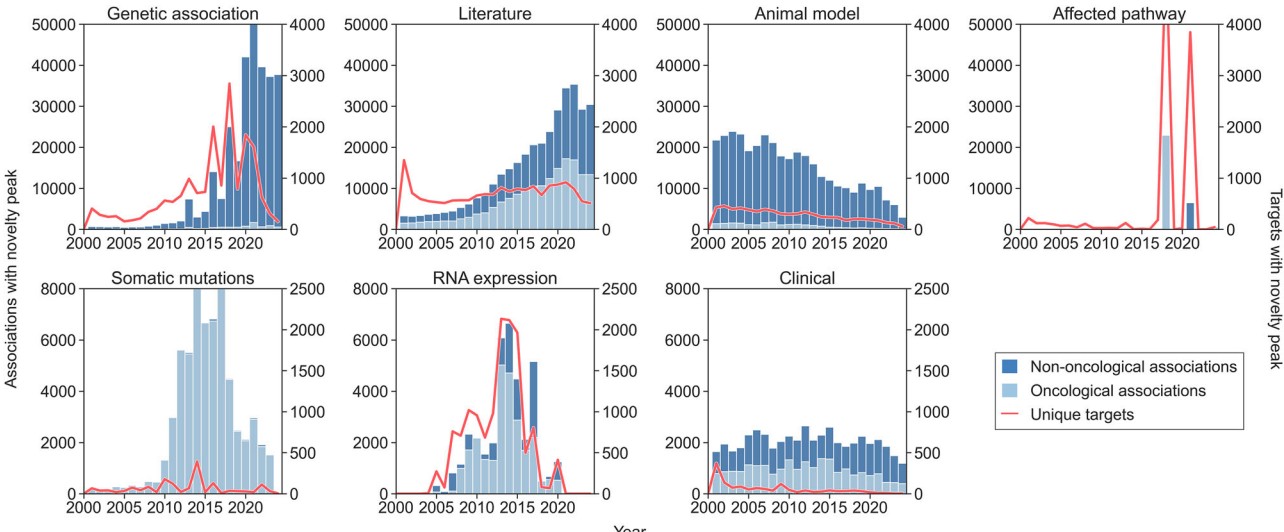

**Fig. 3 | Number of target−disease associations (stacked bar plot) and unique targets (line plot) with novelty peak across data source categories over the years.** Associations are assigned to the year in which the highest novelty peak has been reported in each category. Targets are assigned to the first year in which an association involving them has been reported as novel in each category.

genetic, familial, or congenital diseases (4766). The resources contributing the most associations with novelty peaks are Europe PMC with 43,284 (64%) associations, IMPC with 8997 (13%) associations and GWAS with 6482 (10%) associations.

## Contributions from high-throughput and clinical resources to biomedical novelty

Figure 3 provides a comprehensive analysis of how each Platform resource has contributed to target−disease associations and the identification of novel targets over the past two decades. There has been a striking surge in the number of novel target−disease genetic associations in recent years, reflecting the exponential growth of large-scale genetic studies, and the integration of diverse biobank resources[43,44]. However, this dramatic increase in associations has not been matched by a corresponding rise in unique novel targets. Instead, the majority of recent associations map to DNA regions that were already implicated in previous studies. A similar trend is seen in data extracted from the scientific literature: advances in text mining have led to a rapid increase in the number of reported associations between genes and diseases[45], but the number of unique target genes has remained largely unchanged. This is partly because the research literature tends to focus on already well-known genes, rather than identifying new ones[46]. It is also due to the limitations of current computational frameworks employed to extract biomedical information from text, which often cannot reliably tell the difference between a simple mention of a gene and disease in the same article, and a true, experimentally validated association, such as one supported by evidence of genetic variation or changes in gene expression[47]. Signals of novelty derived from RNA expression resources increased around 2015, corresponding with the increased incorporation of microarray expression studies into the Expression Atlas[48]. The affected pathway resource category shows two peaks of novel association explosion: one in 2018, corresponding to the ingestion of data from the SLA-Penrich analysis[49], which identified significantly mutated pathways in large cancer patient cohorts; and a second one in 2021, corresponding to the ingestion of data from CRISPRbrain: the first genome-wide CRISPR interference and activation screens performed in human neurons[37]. Clinical data shows a related pattern: the number of novel target−disease associations per year has stabilised, while the number of unique new targets entering clinical trials has declined. This suggests that ongoing innovation in clinical research is increasingly

focused on repurposing, new indications, and novel modalities for existing targets rather than introducing first-in-class drugs[16].

## Contributions from expert-curated resources to biomedical novelty

Conversely, expert-curated resources for genetic association evidence (e.g., Gene2Phenotype, Orphanet, GEL PanelApp and ClinGen[50]) offer a closer alignment between the number of novel associations and novel targets discovered over the years. This is despite their modest overall contribution compared to automated methods. Furthermore, multiple curated databases show that they contain similar or identical genetic evidence (see the Supplementary Information). Somatic mutation data, primarily sourced from the Cancer Gene Census (CGC)[51], shows a significant reduction in associations and unique targets over the past decade. This is due to the CGC's recent adoption of a more conservative approach to adding new genes, ensuring the accuracy and reliability of association data[51]. Fig. 3 also reflects a gap between the number of novel associations and novel targets from animal model data, sourced from the International Mouse Phenotyping Consortium (IMPC), similar to that of automated sources. In earlier years, there was a steady influx of new associations and targets as mouse gene knock-out phenotyping progressed. However, in recent years, there has been a progressive decline, suggesting that the resource may be approaching saturation for protein-coding genes[35].

## Retrospective analysis of novel drug targets

To conclude our analysis, we used the retrospectively generated temporal profiles to gain insight into past and current strategies employed to discover novel drug targets. A list of 433 novel drug targets was extracted from ChEMBL by looking up the MoA of drugs approved since 2000. The identified targets were mapped to their earliest approval, the corresponding disease indication, and the year of the highest novelty peak identified in the target−indication association for each resource category. We then retrospectively evaluated the breadth, type and timing of these novelty peaks, in relation to the year of approval. Figure 4 illustrates whether supporting peaks for each resource category typically emerge before (above 0 on the y-axis) or after (below 0 on the y-axis) the first year of drug approval over the years (x-axis). As expected, given the regulatory pathway, novelty peaks from clinical trials, deconvoluted into phases I/II and III, cluster tightly around the time of drug approval. However, for the other

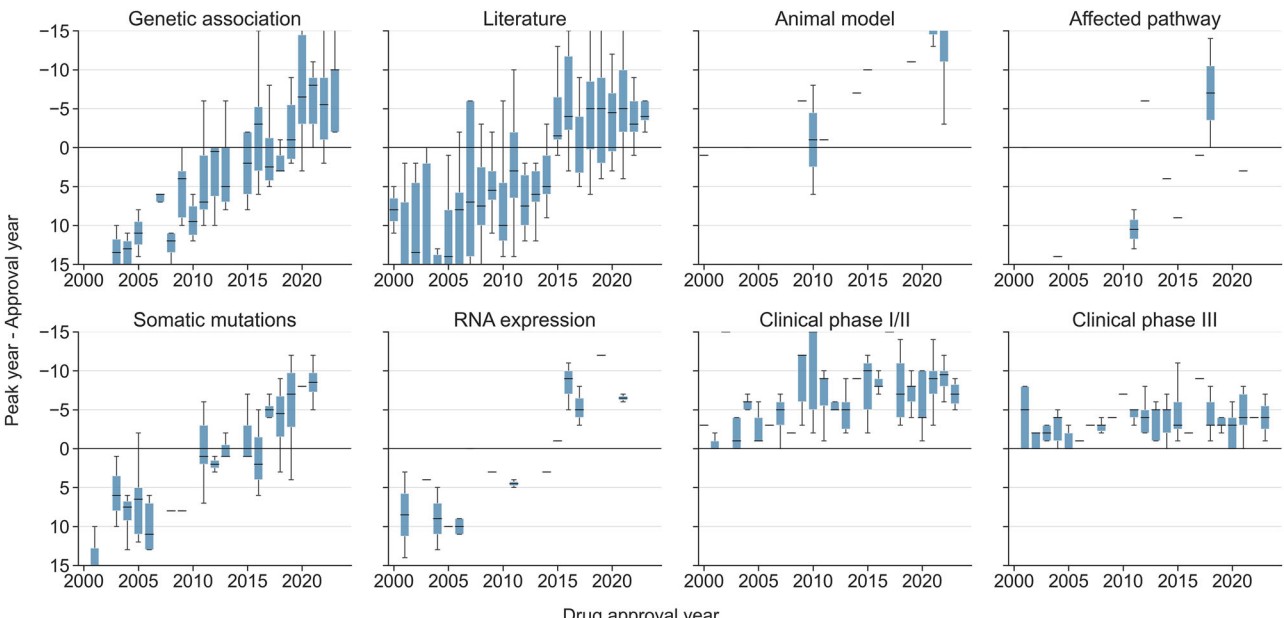

**Fig. 4 | Retrospective analysis of novelty peaks for novel drug targets since 2000.** Each of the 433 novel drug targets (represented by the data points in the box plots) has been mapped to the year of its first drug approval (x-axis), its corresponding disease indication, the year of the highest novelty peak identified in the corresponding target−indication association for each source category, and the number of years elapsed from drug approval to the corresponding top novelty peak (y-axis) for each source category. Clinical peaks have been deconvoluted into Phase I/II and Phase III. These novel drug targets include 302 supported by literature; 72 supported by genetic association; 40 supported by somatic mutation; 26 supported by RNA expression; 16 supported by affected pathway; and 14 supported by animal model evidence. Box plots show the median (centre line), the 25th–75th percentiles (box), whiskers extending to the most extreme points within 1.5 × IQR.

categories we have analysed, we observe that the timing of novelty peaks shifted from occurring after approval to occurring before approval. In all categories except animal models, this shift (the inflection point) took place around 2015. For animal models, the inflection point occurred around 2005. Despite showing greater variability and fewer data points, the affected pathway category also aligns with the general trend. This trend likely arises from a combination of evolving data availability and intentional changes in how this data is utilised. We explore both potential influences in the Discussion section. Furthermore, the percentage of these novel drug targets with biomedical support (whether before or after approval) has remained stable: 70% (302) exhibit peaks in literature novelty; 17% (72) in genetic association novelty; 9% (40) in somatic mutation novelty; 6% (26) in RNA expression novelty; 4% (16) in affected pathway novelty; and 3% (14) in animal model novelty. The only difference is in the timing of these peaks (see the Supplementary Information for more details).

## Discussion

In the post-genome era, advances in high-throughput sequencing and information technologies are dramatically expanding the volume of biomedical data available to help understand disease biology and design better therapies. Despite their huge coverage, there remain challenges in identifying relevant data and interpreting them correctly in order to find evidence that confidently connects diseases with their potential causal gene targets. When considering data mined from text in scholarly literature, patents, and other written sources, large language models especially trained for identifying semantically sound biomedical associations can help bridge this gap by identifying the most relevant articles to prioritise for curation and nominating a preliminary list of associations for expert curation[52]. For GWAS data, translating genetic association signals into individual actionable targets remains challenging in part due to the limited access to summary statistics[44], and this challenge is even greater for less explored genes. Databases providing public access to GWAS data, such as the GWAS Catalogue[43], and open-source frameworks offering post-GWAS

analytics to help predict effector genes, such as Open Targets Gentropy[53], are essential to help pinpoint new causal genes. The importance of human genetic data for successful drug progressability has been explored in numerous publications in recent years[54–58], showing that drug mechanisms with genetic support are 2.6 times more likely to succeed than those without such support[57]. Additionally, up to 47 first-in-class, non-cancer approved drugs have been reported to be directly driven by human genetic observations[59]. In our systematic analysis of 433 novel drug targets with biomedical support for the underlying target−indication association, we found that 23% (101) of them are supported by human genetic data (72 with genetic association evidence and/or 40 with somatic mutation evidence), 70% by literature-derived data, and 13% by other non-clinical biomedical data, with all of these types of evidence increasingly appearing prior to the approval year. We propose two complementary interpretations for this trend. First, most novelty peaks are concentrated within a relatively narrow period, particularly after the emergence of the post-genome era[19], when GWAS, sequencing technologies, and text mining tools became widespread. Second, this trend likely reflects not only the surge in available genomic, transcriptomic, and literature-derived data, but also a growing reliance on such data for the early validation of novel drug targets within the pharmaceutical industry, as discussed by Trajanoska et al. (2023)[59]. As more data are released through public initiatives, some of it retroactively supports previous drug development programmes while also generating new evidence to guide future efforts. This may explain the observed pattern of supporting evidence increasingly emerging before drug approval, suggesting that the industry is shifting towards a greater dependence on publicly available information. Additionally, unlike related studies that propagate supporting evidence through protein interaction networks and disease ontologies[55,56,60], our analysis considers only direct evidence of association between targets and diseases. As a result, our estimates provide a more conservative assessment of supporting evidence. For example, 23% of the novel drug targets have direct human genetic evidence compared to 44% with indirect; 70% with direct literature support

compared to 78% with indirect; and 13% with direct support from other non-clinical sources compared to 52% with indirect. See Supplementary Information for more details. By sharing this temporal analysis, we ultimately hope to facilitate further research in this area and help the scientific community to better understand the evolving role of genetics and other types of biological data in the discovery of novel therapies.

Target selection is a critical decision point in drug discovery. The growing amount of data available that is now relevant to therapeutic target selection and clinical validation makes it increasingly possible to build evidence-based therapeutic hypothesis, but also makes it increasingly challenging for drug discovery scientists to navigate the volume of information for decision-making. Tools such as the Open Targets Platform greatly facilitate this by integrating data from multiple sources and providing public frameworks for analysis. However, as with other open-access resources in this field, it is currently difficult to identify significant changes in the availability of the most relevant data for target−disease associations and assess their novelty. Therefore, in this project, we undertook a comprehensive annotation effort of the 28 million pieces of evidence supporting the 3.6 million target−disease associations in the Open Targets Platform to extract timestamps from each data source, and formulated a new metric to summarise the degree of novelty of a target in the context of a disease according to current available knowledge. The temporal profiles retrospectively constructed for novel drug targets approved over the past two decades suggest an increasing reliance on human genetic, literature-derived, differential expression and pathway-related evidence for target validation throughout the preclinical and clinical pipelines. While these results may be influenced by the tremendous growth in certain areas and types of data over the past decade—genetics being the most obvious example—we anticipate that, in the future, the data and tools we have developed will be invaluable in helping users to navigate the ever-expanding and increasingly complex landscape of life sciences and biomedical data, and to make timely, data-driven decisions about key problems in drug discovery, including which targets to pursue in order to address unmet medical needs.

## Methods

The research work presented in this paper did not require any approval by an ethical committee or institution.

### Biomedical corpus of the Open Targets Platform

The Open Targets Platform biomedical corpus version 25.03 was used in this study and is available at http://ftp.ebi.ac.uk/pub/databases/opentargets/platform/25.03/. It comprises 3.6 million associations between diseases and targets, with supporting evidence derived from over 20 sources (https://platform-docs.opentargets.org/evidence). The pieces of evidence are aggregated by data type into five categories: literature, genetic associations, RNA expression, animal models, somatic mutation, affected pathways and clinical (a.k.a. 'known drugs' which includes approved drugs and clinical candidates). The literature evidence is text-mined from Europe PMC scientific publications and patents. Genetic evidence includes results of GWAS, functional genomic, clinical reports and phenotypic studies curated and deposited into resources such as GEL (Genomics England) PanelApp, Orphanet, ClinVar, Gene2phenotype, Clingen and UniProt; and/or analysed by gene burden studies and Open Targets Genetics. Pieces of evidence from RNA expression experiments are sourced from the Expression Atlas. Genotype-phenotype associations from the International Mouse Phenotypes Consortium (IMPC) are included as animal model evidence. Evidence for cancer mutations and biomarkers is from the Cancer Gene Census, Cancer Genome Interpreter, and a subset of ClinVar that refers to somatic mutation. The ChEMBL team extracts clinical evidence from drug labels, clinical trials and drug approvals that are integrated into the Open Targets ecosystem.

Metabolic pathways involved in pathogenicity identified by systems biology studies and CRISPR screenings are also captured as evidence from projects like Reactome, SLAPenrich, Project Score and CRISPR-brain, and from gene signature publications. A disease or phenotype in the Platform is understood as any disease, phenotype, biological process or measurement that might have any type of causality relationship with a human target. The EMBL-EBI Experimental Factor Ontology (EFO, https://www.ebi.ac.uk/efo/) is used as a scaffold for the disease entity. For a full list of resource references, see Supplementary Information.

### Timestamps of evidence for target−disease associations

A comprehensive timestamping effort was carried out on the 28 million evidence from the Open Targets Platform biomedical corpus (25.03). In order to ascertain the publication and/or submission dates of the evidence, the original sources were consulted. Evidence extracted from Europe PMC documents was annotated with the date of its publication. In the case of resources containing genetic evidence that had been manually curated by experts, the submission date of the curation was annotated. The rationale for this approach is to reduce redundancy in the coverage of evidence from literature sources and curated genetic repositories and to capture the precise moment a given resource becomes aware of a particular piece of evidence when possible. The median time difference between the primary and the curated dates is 11. In the absence of submission dates, the date of publication in the primary source (i.e., a scientific publication) was employed instead. The start year of clinical trials was also recorded. Evidence from pathway-related individual projects were annotated with the project release date or the associated publication. The original resources and links from which the dates were extracted are referenced in the Supplementary Information.

### The Open Targets Platform scoring framework

Every target−disease pair in Open Targets is assigned a harmonised and normalised score that quantifies the strength of the association. This is explained in detail in the Open Targets documentation page (https://platform-docs.opentargets.org/associations). Briefly, the association score is based on the relative importance of the pieces of evidence supporting it and their repetition. While some data sources will capture the meaningful association in a single piece of evidence, in other data sources, the repetition of the evidence increases the confidence with which the association can be regarded as meaningful. To balance all these differences and provide a consensus regarding the strength of the underlying evidence, a harmonisation and normalisation of the scores is performed. Firstly, the evidence is grouped according to the source of origin. Subsequently, data source association scores are calculated by the harmonic sum of the full vector of evidence scores. To ensure the result is between 0 and 1, the harmonic sum is normalised by dividing the result by the maximum theoretical harmonic sum, which is the one calculated using an infinite vector of ones. The Platform derives this calculation (which approximates to 1.644) by using a vector of 1000 ones. Finally, the overall association score is calculated by a second harmonic sum using the vector of data source association scores weighted by the data source weights and normalised in the same way as the source scores.

### The novelty metric formulation

With the evidence annotated with their timestamps, we were able to retrospectively reconstruct the evolution of data source scores for each target−disease pair since 1995. This was achieved by recalculating the scores for each association and each year, considering only evidence accumulated until that time. This resulted in temporal profiles where scores increased as new supporting evidence appeared. Based on these profiles, a metric was defined to quantify the degree of novelty of a target−disease association at a given time. This metric

captures shifts in the score values as peaks of novelty, which subsequently decay as time passes since the shift. In practice, the novelty formula Eq. (1) is defined as a logistic decay function applied to the difference between the score at a given year and the score at previous years, as follows:

$$N = \frac{S}{1 + e^{k(W-m)}} \tag{1}$$

$N$ represents the novelty value at a given year, $S$ is the latest score shift registered, $k$ is the logistic growth rate or steepness of the decay curve, $W$ is the window difference between the current year and the year when the last score shift was registered, and $m$ is the sigmoid decay curve midpoint. A value of 2 was set for the steepness parameter, and a value of 3 was set for the midpoint parameter ad hoc. This allowed for an initial slow decay period in the first and second years after the peak, followed by a faster decay period in the third and fourth years until reaching a zero novelty value again. In the event that several score shifts are registered in consecutive years, all possible novelty values are computed, and the maximum one is selected. In a manner analogous to the overall association score, the overall association novelty is calculated as the harmonic sum of the weighted data source novelty values. A detailed inspection of the number of novelty peaks reported for each target–disease pair has revealed a median value of 1.0 for each data category and maximum values of 3.0 for somatic mutation sources, 5.0 for RNA expression data sources, 6.0 for genetic association and animal model data sources, 8.0 for affected pathway and clinical sources, and 15.0 for literature sources.

### Novelty signals across resources in the Open Targets Platform
The number of target–disease associations and unique targets with novelty signal over the years across resource categories was obtained using a novelty score cutoff of 0.1 to capture more relevant signals. Associations were classified by therapeutic area based on their disease and then assigned to the year in which the highest novelty peak was reported in each category. Targets were assigned to the first year in which an association involving them is reported as novel in each source. No significant changes in the figures were observed when filtering for protein-coding targets only. The following therapeutic areas were excluded: biological process, phenotype, measurement, animal disease and medical procedure.

### Temporal profiles for novel drug targets since 2000
Targets annotated as the mechanism of action of an approved drug according to ChEMBL 34 data were mapped to their first approval and corresponding disease indication. Temporal profiles for the target–disease pairs were recovered, and novelty peaks were subjected to analysis. Highest novelty peaks were selected for each association and source and grouped according to source category. Clinical novelty was evaluated independently of novelty peaks by annotating each target–disease–drug triplet to the earliest clinical trial in the I/II and III phases. A comparative analysis of the temporal patterns of novelty peaks for novel drug targets was conducted, with the data divided into two groups: (a) novel drug targets with their first drug approved between 2000 and 2005, and (b) novel drug targets with their first drug approved between 2020 and 2025. These were selected as the most representative of shifts in the discovery trends of novel drug targets in the last decade.

A large language model-based tool was utilised to assist in refining the clarity and style of selected sections of the manuscript.

### Statistics & reproducibility
No data were excluded from the analyses, and no statistical method was used to predetermine sample size.

### Reporting summary
Further information on research design is available in the Nature Portfolio Reporting Summary linked to this article.

## Data availability
The entire data generated in this study has been deposited on GitHub (https://github.com/opentargets/timeseries) and Zenodo (https://zenodo.org/records/15922783). The source files of biomedical evidence, target and disease data used in this study are available in the Open Targets Platform FTP site: http://ftp.ebi.ac.uk/pub/databases/opentargets/platform/25.03/output/.

## Code availability
The Python code for the current study is publicly available on GitHub: https://github.com/opentargets/timeseries under the following https://doi.org/10.5281/zenodo.17396741.

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

## Acknowledgements

The authors would like to thank our Partners (Wellcome Sanger Institute, EMBL-EBI, Bristol Myers Squibb, Genentech, GSK, MSD, Sanofi and Pfizer) and our Scientific Advisory Board for the crucial strategy discussions. We would especially like to thank Mark McCarthy, Vivek Ramaswamy and the Human Genetics team at Genentech for their insightful discussions and feedback on the manuscript. We also thank Daniel Suveges, Irene Lopez and Annalisa Buniello from the Open Targets team for their help with data access and general feedback, and members of the ChEMBL team, especially Barbara Zdrazil, and the Illuminating the Druggable Genome (IDG) programme, especially Tudor Oprea, for helpful discussions and guidance. This research was partly funded by the European Molecular Biology Laboratory, European Bioinformatics Institute (EMBL-EBI) and Open Targets.

## Author contributions

M.J.F., I.D., A.L., D.O., and E.M. conceived and designed the study. M.J. carried out the methodology, investigation, and visualisation and draughted the manuscript. I.D., A.L., D.O., and E.M. supervised the study. All authors, including D.H., J.M.R., and P.R. aided in the interpretation of data and/or critical revision of the manuscript.

## Funding

## Competing interests

The authors declare no competing interests.
