## [Transparent Peer Review file · Nature Communications]

Temporal trends in evidence supporting novel drug target discovery

Corresponding Author: Dr Maria Falaguera

Version 0:

Reviewer comments:

Reviewer #1

(Remarks to the Author)

The identification of novel target-indication pairs is a seminal problem in drug discovery. OpenTargets is the premier resource in this space and is used widely. This paper discusses the OpenTargets (OTs) effort to time-stamp the origin of the evidence across multiple data resources. This is a valuable endeavor as it can quantitatively identify novel target-indication pairs. However, the key conclusions of the paper need more support or need to be restated.

1. The analysis is for target-indication pairs but the introductory stats in the first paragraph lines 37-44 appear to be for novel targets.
2. Shouldn't regulatory approval for a single-target monoclonal lead to a perfect evidence score for that indication (lines 158-159)?
3. Consider assessing what fraction of literature-based evidence is in Europe PMC. If the fraction is small then perhaps the timestamp from Europe PMC is not as useful.
4. Consider leaving out half of the data from each source to see how it affects your results and conclusions.
5. Figure 1b: consider showing a different plot that shows the density of the data for each year. It is likely that a large fraction of the data is of recent vintage, thus making your task harder and might be biasing your conclusions.
6. Figure 2b and the explanation in lines 134-161 is a helpful exposition of your methodology.
Line 196: Are novelty scores a good way to assess the quantity of evidence? I thought the OT evidence score was better for that purpose.
7. Line 236, do you need to correct for the amount of evidence each year?
8. Figure 3c: please explain that the size/area of each is proportional to its novelty score in 2023. What is the lowest shown score? This is useful data and I think the critical value of your paper.
9. Figure 4: One possible explanation (and perhaps the simplest one) is that it shows that the evidence is mostly in the middle years and the trend shows nothing significant. For the shifting of peaks to be significant needs more analysis/evidence with better controls and
10. The key explanation for the title appears to be in lines 300-9. But I just don't see how these lines support the title "Temporal trends in novel drug target discovery reveal the increasing importance of human genetic data". If anything one could argue that the "Temporal trends in novel drug target discovery reveal the increasing importance of ALL data". However, I would suggest something along the lines of "Quantifying Novelty of Target Indication Pairs ..."

Pankaj Agarwal

(Remarks on code availability)

Reviewer #2

(Remarks to the Author)

Analysis of Research Paper: Temporal Trends in Novel Drug Target Discovery

Noteworthy Results

What are the noteworthy results?

- Methods and public implementations demonstrating how to best temporalize biomedical evidence
- This is crucial for retrospective machine learning methods, and in my opinion erroneously omitted many prior ML-based studies using such evidence to predict clinical outcomes.
- Descriptive statistics on trends in biomedical evidence at a deeper level than targets
- This a good complement to prior work like [Nájera et al. 2021](<https://www.nature.com/articles/s41598-021-94897-9>), and similar work attempting to quantify trends for targets, in that trends at a deeper level like target-disease pairs are more informative for many applications.
- Characterizing the recency with which evidence has arisen in sources across Open Targets (Figure 1)
- This is likely to be a very helpful resource for many users of the platform.
- A demonstration of how evidence prevalence for targets post-approval increases across nearly all sources (Figure 4)
- This provides much needed quantitative evidence on the extent to which clinical success likely drives future discovery.
- The increasing prevalence of novelty peaks in genetic evidence prior to approval in more recent years (Figure 5)
- This is a great caveat, however: "Whether these trends are driven by the explosion of genetic data over the past decade or reflect the increasing reliance on genetics for early validation of novel drug targets remains to be seen".
- Consider citing [Trajanoska et al. 2023](<https://www.nature.com/articles/s41586-023-06388-8>) for this. That it is probably the best study I have personally seen at providing more likely causal evidence linking increasing genetic evidence to the initiation of new drug programs. It would further the argument that the trends observed in this study are not purely correlative.
- The disparity in novel discoveries within rare diseases and other therapeutic areas

Significance to the Field

Will the work be of significance to the field and related fields? How does it compare to the established literature? If the work is not original, please provide relevant references.

- This work is of very high significance to the field in my opinion, as it enables retrospective studies at a level beyond targets with more rigor
- In particular, careful strategies for temporalizing evidence by source like this are very valuable:
<https://github.com/opentargets/timeseries/blob/48851fdace904dc037a8f4f74e14503952291eab/timestampsParser.py>.
- I'm not aware of any other studies on trends within target-disease associations across many sources like this.
- The closest I know of is our work at <https://doi.org/10.1101/2024.08.02.24311422>.

Data Analysis, Interpretation, and Conclusions

Does the work support the conclusions and claims, or is additional evidence needed? Are there any flaws in the data analysis, interpretation and conclusions? Do these prohibit publication or require revision?

- As a largely descriptive study on trends, I believe this manuscript does a great job of highlighting noteworthy patterns without making strong claims about their origins .
- I see no need for additional evidence.
- I think this is concerning:
<https://github.com/opentargets/timeseries/blob/48851fdace904dc037a8f4f74e14503952291eab/timeseries.py#L961-L1022>.
- The use of indirect evidence will make many of the novelty signals highly redundant, i.e. a parent term will have equivalent novelty scores to a child term within EFO if there are no "peaks" for other children.
- This means that aggregations of the novelty score up to targets or therapeutic areas (TAs) are highly dependent up the structure of EFO for those entities, in particular the depth of the subtree descending from them.
- This also calls the claim that rare diseases are depleted for novelty into question since their EFO trees are much shallower than other TAs
- If I'm reading the code correctly, here's an example of the problem:
- Consider [Evidence for BMP2 in heritable pulmonary arterial hypertension] (https://platform.opentargets.org/evidence/ENSG00000204217/MONDO_0017148)
- This disease has an EFO path like: hypertension -> pulmonary hypertension -> pulmonary arterial hypertension -> heritable pulmonary arterial hypertension.
- Assume a novelty peak occurs for BMP2 and heritable pulmonary arterial hypertension due to a genetic association.
- Now the associations count in a result like Figure 3 for the cardiovascular TA should include increment by at least 4 because of this single new genetic association.

- One solution would be to count genes instead of associations for Figure 3, which I see is readily available in [analysis.ipynb] (<https://github.com/opentargets/timeseries/blob/48851fdace904dc037a8f4f74e14503952291eab/analysis.ipynb>).
- Another solution would be to use direct evidence.

Methodology

Is the methodology sound? Does the work meet the expected standards in your field?

I believe the methodology is sound. The implementation of that methodology also appears sound. I have reviewed much of <https://github.com/opentargets/timeseries> and see no obvious issues.

Reproducibility

Is there enough detail provided in the methods for the work to be reproduced?

Yes, there is enough detail in the provided code to reproduce the study.

Recommendations

What improvements or revisions could be suggested?

- On date selection strategies:
 - The difference between a "project date" and a "primary publication date" is not very clear. That could use some further detail.
 - How far off can either the curation, project or publication date be from the true discovery date? Some discussion on this could also be useful.
- On figures:
 - Figure 5
 - A useful expansion on this would examine novelty peaks prior to early clinical development rather than approval.
 - This would be suggestive of how often evidence substantiates investment in those programs (cf. Trajanoska et al. 2023) rather than possibly being derived directly or indirectly as a result of them.
 - Figure 4
 - It would be helpful to understand how much of the novelty signals following approval for a target are for **different** indications than those originally approved.
- On applications of this metric:
 - I would personally like to see this expanded on: "we anticipate that in the future the data and tools we have developed will prove invaluable in helping users navigate the ever-expanding and increasingly complex landscape of life sciences and biomedical data to make timely, data-driven decisions about key problems in drug discovery"
 - For what applications would these users apply this novelty metric?
 - It would be helpful to see some discussion or evidence demonstrating how this metric enriches for future clinical development across targets or target-disease pairs.
 - Enrichment for approval or clinical success would also further substantiate the value of the metric, even though the connection between novelty and success is less clear. There is a connection though in that the novelty metric will likely correlate with reproducibility of discoveries, so it may also correlate with clinical success.

Conclusion

There are no critical issues I see with this study other than the use of indirect evidence in some results/figures. It's excellent work and I'm excited to see more like it in the future!

****Reviewer****: Eric Czech

(Remarks on code availability)

I have reviewed the code as mentioned in my comments to the authors. I did not run it myself, but I am quite familiar with this technology stack, their previous codebases and what is necessary to reproduce something they have done. I see no reason to believe this study couldn't be reproduced.

Reviewer #3

(Remarks to the Author)

The authors describe the development of a time-stamp for the emergence different types of evidence available through the Open Targets platform, the creation of a metric for novelty, and several analyses investigating the temporal relationship between target novelty and approval. This work provides a valuable addition to the Platform and this paper provides both documentation for and some initial exploratory analyses of the novelty metric. There are a variety of questions that researchers can use this resource to investigate further.

I have no major concerns with the paper. Below are minor comments and questions I had as I carefully read the manuscript and the accompanying materials.

- Page 4: “phase IV clinical trials are initiated” – as recently noted by Open Targets in the most recent improvement in the use of ChEMBL data, phase 4 trial status provides no added information about the suitability of a drug mechanism for a particular indication. I suggest this manuscript be updated to reflect that improved view of the ChEMBL evidence.
- Figure 2c: As described in the methods, you calculate an overall association novelty across sources. It would be helpful to include this score in the example to illustrate this.
- Page 7: “greater than 0.4, a threshold for high novelty scores (see Supplementary Information for more details).” I could not find any details about this in the supplementary information, aside from Table S3, which is not referenced here.
- Fig. 3a: there is no scale provided for the heatmap. Are these relative from min to max on a linear scale? Please add to the figure or describe in the legend.
- Fig. 3c: Please replace the word cloud with some other form of visual representation, or remove altogether and refer to the supplementary table. I am unable to reconcile the differences between the csv file provided and this figure. Based on the table, CHEK2 and several other genes have higher novelty scores than those shown in this figure.
- Fig. 4: The linear increases in the median differences from peak year to approval year suggests that for most of these evidence sources, most of the peaks occurred in a relatively narrow timeframe. Could you explore this further? E.g. for each source of evidence, the count of peaks by year as you did in Fig. 3b. Probably with and without restrictions to the targets of approved drugs. I believe that this figure is likely to be more informative than the current Fig. 5, so depending on the patterns, you may want to swap that and move Fig. 5 to supplement.
- Data availability: I went to the specified GitHub site and found the code, but not the the underlying data. If code is required to pull the data from an online location, this was not clear. I was expecting to find one or more files that contained the novelty peaks for each gene-trait-year-evidence type, similar to the 2023 example file provided.

I hope you find this review helpful.

Warmest regards,
Matthew R. Nelson

(Remarks on code availability)

Version 1:

Reviewer comments:

Reviewer #1

(Remarks to the Author)

The authors have adequately answered previous comments.

(Remarks on code availability)

Reviewer #2

(Remarks to the Author)

All of my initial concerns have been appropriately addressed by the authors.

(Remarks on code availability)

Reviewer #3

(Remarks to the Author)

The authors have addressed the concerns I raised in the original review and have substantially improved the overall manuscript in addressing the comments of the other reviewers. I have no further comments.

(Remarks on code availability)

Reviewer #1 (Remarks to the Author):

The identification of novel target-indication pairs is a seminal problem in drug discovery. OpenTargets is the premier resource in this space and is used widely. This paper discusses the OpenTargets (OTs) effort to time-stamp the origin of the evidence across multiple data resources. This is a valuable endeavor as it can quantitatively identify novel target-indication pairs. However, the key conclusions of the paper need more support or need to be restated.

1. The analysis is for target-indication pairs but the introductory stats in the first paragraph lines 37-44 appear to be for novel targets.

Thank you for your comment and for drawing attention to this point. The statistics presented in the Introduction are indeed based on novel mechanism of action targets of approved drugs, which correspond to drug–target–indication triplets. The novelty metric introduced in this manuscript serves a dual purpose: (i) to identify novel targets in the context of a specific disease, and (ii) to retrospectively analyse the novelty peaks associated with target–indication associations at the time of first drug approval with that target as its mechanism of action. To make this clearer, in the section “Retrospective analysis of novel drug targets” of the revised manuscript we now state: “We map each novel drug target to its year of first drug approval, its corresponding disease indication, and the year of the highest novelty peak identified in the corresponding target–indication association for each category source.” We hope this clarification addresses your concern and improves the alignment between the introductory statistics and the main analysis.

2. Shouldn't regulatory approval for a single-target monoclonal lead to a perfect evidence score for that indication (lines 158-159)?

Thank you for this insightful question. According to the Open Targets Platform documentation (<https://platform-docs.opentargets.org/evidence/evidence-scoring>), “the repetition of evidence increases our confidence in the association. The scoring by data source provides a consensus view on the strength of the underlying evidence for a particular source.” While regulatory approval of a single-target monoclonal antibody does yield a perfect ChEMBL evidence score of 1.0 for that specific drug, the harmonised ChEMBL source score is designed to reflect cumulative confidence. This means that target–indication associations supported by multiple independent drug approvals will accrue higher confidence scores than those with only one approval, even if that approval is for a single-target monoclonal. To date, this approach has not raised concerns within the Open Targets Platform community (<https://community.opentargets.org/>). However, we appreciate your suggestion and encourage further discussion of this scoring methodology within the community to ensure it continues to meet user needs. We have clarified this point in “The Open Targets Platform scoring framework” within the Methods section in the revised manuscript.

3. Consider assessing what fraction of literature-based evidence is in Europe PMC. If the fraction is small then perhaps the timestamp from Europe PMC is not as useful.

According to recent publications (<https://pmc.ncbi.nlm.nih.gov/articles/PMC10767826/>, <https://blog.europepmc.org/2025/05/making-sense-of-europe-pmc-answers-to-your-biggest-faqs.html>), Europe PMC provides comprehensive coverage of life sciences literature, including all

journal article abstracts from PubMed/MEDLINE and full text articles from PubMed Central (PMC), as well as preprints from over 30 servers, books, and theses. This means that the vast majority of literature-based evidence indexed in major biomedical databases is also present in Europe PMC, making it essentially a mirror of PubMed and PMC, with only a small number of historical articles predating 2006 not included. Despite some associations might be missing through our literature extraction pipeline (<https://academic.oup.com/bioinformatics/article/41/4/btaf113/8082101>) we are confident it is comprehensive enough.

4. Consider leaving out half of the data from each source to see how it affects your results and conclusions.

Thank you for this suggestion. In response to feedback from several reviewers, we have now implemented a more stringent data selection in the revised manuscript by considering only “direct” evidence linking targets and diseases, as opposed to the previous version where “indirect” evidence (propagated across the disease ontology) was included. This adjustment resulted in a substantial reduction in the number of data points, arguably an even more stringent test than leaving out half of the data from each source. Importantly, despite this significant reduction, our main results and conclusions—particularly those presented in Figure 3 and Figure 4—remain unchanged. We believe this demonstrates the robustness of our findings and hope this addresses the reviewer’s concerns.

5. Figure 1b: consider showing a different plot that shows the density of the data for each year. It is likely that a large fraction of the data is of recent vintage, thus making your task harder and might be biasing your conclusions.

Thank you for this suggestion. In response to feedback from several reviewers, we have now assessed and visualised the density of data over time in the revised manuscript, specifically in the new Figure 3. In addition, we discuss the implications of the temporal distribution and potential biases arising from this in subsections “Contributions from high-throughput and clinical resources to biomedical novelty” and “Contributions from expert-curated resources to biomedical novelty” in the Results section, and in the Discussion section.

6. Figure 2b and the explanation in lines 134-161 is a helpful exposition of your methodology. Line 196: Are novelty scores a good way to assess the quantity of evidence? I thought the OT evidence score was better for that purpose.

Thank you for your comment and for highlighting this important distinction. According to the Open Targets Platform documentation (<https://platform-docs.opentargets.org/evidence/evidence-scoring>), the OT evidence score is well-suited for assessing the confidence in individual pieces of evidence supporting a target–disease association. The association score aggregates these evidence scores across sources, providing a robust measure of both the quantity and quality of supporting evidence. The association novelty metric introduced in our study adds a temporal perspective, allowing us to track how the onset, quality, and quantity of evidence for an association evolve over time. To clarify this in the manuscript, we have rephrased the sentence on line 196 as follows: “In summary, Figure 2 shows the differences between the accumulation of evidence for a target–disease association and the evolution of the Open Targets Platform association and

novelty scores, with novelty peaks providing a 'clear' view of the onset, quality and quantity of evidence over time." We hope this revision addresses your concern and more accurately reflects the complementary roles of these metrics in our analysis.

7. Line 236, do you need to correct for the amount of evidence each year?

In response to feedback from several reviewers, we have revised Figure 3 and updated the corresponding descriptions in the Results and Discussion sections (where the former line 236 was included) to better account for the variation in the amount of evidence available each year. These revisions allow for a more accurate interpretation of temporal trends and help address potential biases arising from fluctuations in yearly evidence volume. We hope these changes adequately address your concern.

8. Figure 3c: please explain that the size/area of each is proportional to its novelty score in 2023. What is the lowest shown score? This is useful data and I think the critical value of your paper.

Thank you very much for your insightful comment. We appreciate that Figure 3c and the associated word cloud were considered a critical part of the manuscript. However, in the revised version, we have updated our analysis to use data from a later release of the Open Targets Platform (25.03), which includes broader and more current data. This update changed the list of novel targets, making the previous word cloud from release 23.06 no longer representative. To maintain the robustness and relevance of our presentation, we have removed the original Figure 3c and replaced it with a new Figure 3 that provides a more general and atemporal overview of novel targets. This figure captures a three-dimensional concept: the number of target–disease associations and unique targets identified as novel over the years, stratified by oncological and non-oncological and by data categories. We believe this approach better reflects the evolving nature of the data and avoids bias introduced by release-specific snapshots. We have preserved the detailed list of novel targets for the new release as Supplementary Information to ensure transparency and accessibility. We hope this revised presentation addresses your concerns while strengthening the manuscript's scientific rigor and longevity. Thank you again for your valuable feedback, which has helped us improve the clarity and impact of our work.

9. Figure 4: One possible explanation (and perhaps the simplest one) is that it shows that the evidence is mostly in the middle years and the trend shows nothing significant. For the shifting of peaks to be significant needs more analysis/evidence with better controls and

Thank you for this valuable observation. In response, we have revised the Results and Discussion sections related to Figure 4 to provide a more cautious and nuanced interpretation. We now explicitly acknowledge the possibility that the observed trend may simply reflect the concentration of evidence in the middle years, as the reviewer suggests, and that the shifting of peaks may not, by itself, indicate a significant underlying phenomenon without further analysis and appropriate controls. Specifically, in the "Retrospective analysis of novel drug targets" subsection in the Results section, and in the Discussion section, we have added the following clarification: "We propose two complementary interpretations for this trend. First, most novelty peaks are concentrated within a relatively narrow period, particularly after the emergence of the post-genome era,¹⁹ when GWAS, sequencing technologies, and text mining tools became widespread. Second, this trend likely

reflects not only the surge in available genomic, transcriptomic, and literature-derived data, but also a growing reliance on such data for the early validation of novel drug targets within the pharmaceutical industry, as discussed by Trajanoska et al. (2023).⁵⁹ [...]". We hope this revision addresses your concern and provides a more balanced interpretation of the data. Thank you again for your constructive feedback.

10. The key explanation for the title appears to be in lines 300-9. But I just don't see how these lines support the title "Temporal trends in novel drug target discovery reveal the increasing importance of human genetic data". If anything, one could argue that the "Temporal trends in novel drug target discovery reveal the increasing importance of ALL data". However, I would suggest something along the lines of "Quantifying Novelty of Target Indication Pairs ..."

Thank you for this insightful comment. We agree that the previous title may have overstated the specific contribution of human genetic data, as the temporal trends observed in our analysis reflect the increasing importance of multiple types of supporting evidence. In response to your suggestion, we have revised the manuscript title to: "Temporal trends in evidence supporting novel drug target discovery" We believe this new title more accurately reflects the main findings and general trend highlighted by the reviewer, while preserving the core focus of our research. Thank you again for helping us improve the clarity and relevance of our manuscript.

Pankaj Agarwal (pa@Bioinfi.com)

Reviewer #2 (Remarks to the Author):

Analysis of Research Paper: Temporal Trends in Novel Drug Target Discovery

Noteworthy Results

What are the noteworthy results?

- Methods and public implementations demonstrating how to best temporalize biomedical evidence
- This is crucial for retrospective machine learning methods, and in my opinion erroneously omitted many prior ML-based studies using such evidence to predict clinical outcomes.
- Descriptive statistics on trends in biomedical evidence at a deeper level than targets
- This a good complement to prior work like [Nájera et al. 2021](<https://www.nature.com/articles/s41598-021-94897-9>), and similar work attempting to quantify trends for targets, in that trends at a deeper level like target-disease pairs are more informative for many applications.
- Characterizing the recency with which evidence has arisen in sources across Open Targets (Figure 1)
- This is likely to be a very helpful resource for many users of the platform.
- A demonstration of how evidence prevalence for targets post-approval increases across nearly all sources (Figure 4)
- This provides much needed quantitative evidence on the extent to which clinical success likely drives future discovery.
- The increasing prevalence of novelty peaks in genetic evidence prior to approval in more recent years (Figure 5)
- This is a great caveat, however: "Whether these trends are driven by the explosion of genetic data over the past decade or reflect the increasing reliance on genetics for early validation of novel drug targets remains to be seen".
- Consider citing [Trajanoska et al. 2023](<https://www.nature.com/articles/s41586-023-06388-8>) for this. That it is probably the best study I have personally seen at providing more likely causal evidence linking increasing genetic evidence to the initiation of new drug programs. It would further the argument that the trends observed in this study are not purely correlative.
- The disparity in novel discoveries within rare diseases and other therapeutic areas

Thank you for highlighting this important caveat and for recommending Trajanoska et al. (2023) as a key reference. We agree that the question of whether observed trends are due to the explosion of genetic data or reflect an increasing reliance on genetics for early drug target validation is central to interpreting our findings. To strengthen this aspect of our manuscript, we have explicitly acknowledged the caveat in the Results and Discussion sections and cited Trajanoska et al. (2023) in that context. Specifically, in the "Retrospective analysis of novel drug targets" subsection in the Results section, and in the Discussion section, we have added the following clarification: "We propose two complementary interpretations for this trend. First, most novelty peaks are concentrated within a relatively narrow period, particularly after the emergence of the post-genome era,¹⁹ when GWAS, sequencing technologies, and text mining tools became widespread. Second, this trend likely reflects not only the surge in available genomic, transcriptomic, and literature-derived data, but also a growing reliance on such data for the early validation of novel drug targets within the pharmaceutical

industry, as discussed by Trajanoska et al. (2023).⁵⁹ [...]". We hope this revision addresses your concern and provides a more balanced interpretation of the data. Thank you again for your constructive feedback.

Significance to the Field

Will the work be of significance to the field and related fields? How does it compare to the established literature? If the work is not original, please provide relevant references.

- This work is of very high significance to the field in my opinion, as it enables retrospective studies at a level beyond targets with more rigor
- In particular, careful strategies for temporalizing evidence by source like this are very valuable: <https://github.com/opentargets/timeseries/blob/48851fdace904dc037a8f4f74e14503952291eab/timestampsParser.py>.
- I'm not aware of any other studies on trends within target-disease associations across many sources like this.
- The closest I know of is our work at <https://doi.org/10.1101/2024.08.02.24311422>.

Thank you for the comment. We have now added a citation to this preprint in the Discussion section.

Data Analysis, Interpretation, and Conclusions

Does the work support the conclusions and claims, or is additional evidence needed? Are there any flaws in the data analysis, interpretation and conclusions? Do these prohibit publication or require revision?

- As a largely descriptive study on trends, I believe this manuscript does a great job of highlighting noteworthy patterns without making strong claims about their origins .
- I see no need for additional evidence.
- I think this is concerning: <https://github.com/opentargets/timeseries/blob/48851fdace904dc037a8f4f74e14503952291eab/timeseries.py#L961-L1022>.
- The use of indirect evidence will make many of the novelty signals highly redundant, i.e. a parent term will have equivalent novelty scores to a child term within EFO if there are no "peaks" for other children.
- This means that aggregations of the novelty score up to targets or therapeutic areas (TAs) are highly dependent up the structure of EFO for those entities, in particular the depth of the subtree descending from them.
- This also calls the claim that rare diseases are depleted for novelty into question since their EFO trees are much shallower than other TAs
- If I'm reading the code correctly, here's an example of the problem:
- Consider [Evidence for BMP2 in heritable pulmonary arterial hypertension](https://platform.opentargets.org/evidence/ENSG00000204217/MONDO_0017148)
- This disease has an EFO path like: hypertension -> pulmonary hypertension -> pulmonary arterial hypertension -> heritable pulmonary arterial hypertension.

- Assume a novelty peak occurs for BMP2 and heritable pulmonary arterial hypertension due to a genetic association.
- Now the associations count in a result like Figure 3 for the cardiovascular TA should include increment by at least 4 because of this single new genetic association.
- One solution would be to count genes instead of associations for Figure 3, which I see is readily available in [\[analysis.ipynb\]\(https://github.com/opentargets/timeseries/blob/48851fdace904dc037a8f4f74e14503952291eab/analysis.ipynb\)](https://github.com/opentargets/timeseries/blob/48851fdace904dc037a8f4f74e14503952291eab/analysis.ipynb).
- Another solution would be to use direct evidence.

Thank you for this suggestion. In response to feedback from several reviewers, in the reviewed version we have now implemented a more stringent data selection in the revised manuscript by considering only direct evidence linking targets and diseases, as opposed to the previous version where indirect evidence (propagated across the disease ontology) was included. This adjustment resulted in a substantial reduction in the number of data points as it can be appreciated in all the figures and general counts. Importantly, despite this significant reduction, our main results and conclusions—particularly those presented in Figure 3 and Figure 4 (and former Figure 5)—remain unchanged. We believe this demonstrates the robustness of our findings and hope this addresses the reviewer’s concerns. Furthermore, Figure 3 has been modified to capture a three-dimensional concept: the number of target–disease associations and unique targets identified as novel over the years, stratified by oncological and non-oncological and by data categories. We believe this approach better reflects the evolving nature of the data and avoids bias introduced by release-specific snapshots, as was the case with the previous version of Figure 3. We have preserved the detailed list of novel targets for the new release as Supplementary Information to ensure transparency and accessibility. The figures accounting for indirect evidence are provided as Supplementary Information too.

Methodology

Is the methodology sound? Does the work meet the expected standards in your field?

I believe the methodology is sound. The implementation of that methodology also appears sound. I have reviewed much of <https://github.com/opentargets/timeseries> and see no obvious issues.

Reproducibility

Is there enough detail provided in the methods for the work to be reproduced?

Yes, there is enough detail in the provided code to reproduce the study.

Recommendations

What improvements or revisions could be suggested?

- On data selection strategies:

- *The difference between a "project date" and a "primary publication date" is not very clear. That could use some further detail.*

Thank you for this comment. We agree that there is no significant difference between project date and primary publication date since the project date is annotated with the date of the primary publication presenting the project. In the reviewed version only "Primary publication date" and "Curation date" are distinguished both in Figure 1b, in "Timestamping evidence supporting target–disease associations" subsection in the Results section, and in "Timestamps of evidence for target–disease associations" subsection in the Methods section. We hope this clarification addresses your concern.

- *How far off can either the curation, project or publication date be from the true discovery date? Some discussion on this could also be useful.*

Thank you for raising this important point. As indicated in the Methods section, there is a median lag of 11 years between the primary publication date and the curation date—that is, the time between when genetic evidence first appears in the literature and when it is formally submitted by an expert curator to a resource. This lag reflects the time required for evidence to be recognised, curated, and integrated into databases. Some of these resources are newer but will curate older literature as well as new literature hence the 11-years-gap estimated. Despite the limitations of this assumption, we treat the date of the primary publication of evidence as an approximation of the true date of discovery. We hope this clarifies the potential discrepancies between discovery, publication, and curation dates, and addresses your concern.

- *On figures:*

- *Figure 5*

- *A useful expansion on this would examine novelty peaks prior to early clinical development rather than approval.*

- *This would be suggestive of how often evidence substantiates investment in those programs (cf. Trajanoska et al. 2023) rather than possibly being derived directly or indirectly as a result of them.*

Thank you for this thoughtful and valuable suggestion. In the revised manuscript, we have expanded our analysis by adding a figure to the Supplementary Information that mirrors Figure 5, comparing the novelty peak year relative to the initiation of the first Phase I or II clinical trial for the novel drug target–indication association. While we appreciate the importance of this perspective, we have chosen not to include additional discussion of this figure in the main text, in order to avoid overloading the manuscript and to maintain a clear and focused narrative. We hope you understand our decision and appreciate the scope of the current work.

- *Figure 4*

- *It would be helpful to understand how much of the novelty signals following approval for a target are for ****different**** indications than those originally approved.*

Thank you for this insightful comment. In our analysis, novelty signals shown in Figure 4 are exclusively tied to the original disease indication for which each target was first approved, based

on its mechanism of action as reported in ChEMBL. We do not consider novelty signals associated with subsequent or different indications for the same target. We have clarified this in the revised manuscript in the “Retrospective analysis of novel drug targets” Results subsection with the following paragraph: “A list of 433 novel drug targets has been extracted from ChEMBL by looking up for the MoA of drugs approved since 2000. The identified targets have been mapped to their earliest approval, the corresponding disease indication, and the year of the highest novelty peak identified in the target–indication association for each resource category. Then, we have retrospectively evaluated the breadth, type and timing of these novelty peaks, in relation to the year of approval”. Also, we have further clarified this in the “Temporal profiles for novel drug targets since 2000” Methods’ subsection with the following paragraph: “Targets annotated as the mechanism of action of an approved drug according to ChEMBL 35 data were mapped to their first approval and corresponding disease indication. Temporal profiles for the target–disease pairs were recovered and novelty peaks were subjected to analysis. Highest novelty peaks were selected for each association and source and grouped according to source category.”. We hope this addressed the reviewers’ question.

- On applications of this metric:

- I would personally like to see this expanded on: “we anticipate that in the future the data and tools we have developed will prove invaluable in helping users navigate the ever-expanding and increasingly complex landscape of life sciences and biomedical data to make timely, data-driven decisions about key problems in drug discovery”. For what applications would these users apply this novelty metric? It would be helpful to see some discussion or evidence demonstrating how this metric enriches for future clinical development across targets or target-disease pairs. Enrichment for approval or clinical success would also further substantiate the value of the metric, even though the connection between novelty and success is less clear. There is a connection though in that the novelty metric will likely correlate with reproducibility of discoveries, so it may also correlate with clinical success.

Thank you for these insightful comments. We agree that exploring the downstream applications of the novelty metric—such as its potential to enrich future clinical development or approval—would be highly valuable. However, as this would require a substantial additional analysis beyond the current scope, we have opted not to extend the manuscript further in order to maintain a clear focus on the development of the metric and its use to characterise trends and patterns in biomedical data. That said, we do envision future work exploring these important applications in more depth, particularly in relation to reproducibility, translational potential, and success in clinical pipelines. We hope the reviewer understands this decision and appreciates the current contribution as a foundation for such future analyses.

Conclusion

There are no critical issues I see with this study other than the use of indirect evidence in some results/figures. It's excellent work and I'm excited to see more like it in the future!

****Reviewer**:** Eric Czech

Reviewer #2 (Remarks on code availability):

I have reviewed the code as mentioned in my comments to the authors. I did not run it myself, but I am quite familiar with this technology stack, their previous codebases and what is necessary to reproduce something they have done. I see no reason to believe this study couldn't be reproduced.

Reviewer #3 (Remarks to the Author):

The authors describe the development of a time-stamp for the emergence different types of evidence available through the Open Targets platform, the creation of a metric for novelty, and several analyses investigating the temporal relationship between target novelty and approval. This work provides a valuable addition to the Platform and this paper provides both documentation for and some initial exploratory analyses of the novelty metric. There are a variety of questions that researchers can use this resource to investigate further.

I have no major concerns with the paper. Below are minor comments and questions I had as I carefully read the manuscript and the accompanying materials.

• Page 4: "phase IV clinical trials are initiated" – as recently noted by Open Targets in the most recent improvement in the use of ChEMBL data, phase 4 trial status provides no added information about the suitability of a drug mechanism for a particular indication. I suggest this manuscript be updated to reflect that improved view of the ChEMBL evidence.

Thank you for this important comment. We agree with the updated view recently highlighted by Open Targets that Phase IV clinical trial status does not provide additional insight into the suitability of a drug mechanism for a specific indication, as it typically reflects post-marketing surveillance rather than new evidence of efficacy. In response, we have revised the "Temporal profiles for target–disease associations" subsection in the Results to focus on pre–Phase IV clinical trials. We hope this revision fully addresses the reviewer's concern.

• Figure 2c: As described in the methods, you calculate an overall association novelty across sources. It would be helpful to include this score in the example to illustrate this.

Thank you for this suggestion. In the revised manuscript, we have updated Figure 2a to include the overall association score and novelty and commented it in the "Temporal profiles for target–disease associations" subsection in Results: "Aggregating, harmonising and normalising the three source association scores produces an overall association score curve showing two main shifts".

• Page 7: "greater than 0.4, a threshold for high novelty scores (see Supplementary Information for more details)." I could not find any details about this in the supplementary information, aside from Table S3, which is not referenced here.

*Thank you for raising this point. In the revised manuscript, we have updated the sections analysing novelty signals for 2025 (formerly 2023) and across the years. The counts of novel associations and unique targets reported in the new section "Biomedical associations with novelty signals in 2025" are based on applying a novelty threshold of 0.1 instead of 0.4. This less restrictive, *ad hoc* threshold was chosen to reduce noise arising primarily from very low-confidence evidence, largely sourced from Europe PMC. This new cutoff is now consistently applied throughout the entire*

Results section and indicated in the “Novelty signals across resources in the Open Targets Platform” section in Methods with the following paragraph: *“The number of target–disease associations and unique targets with novelty signal over the years across resources categories were obtained using a novelty score cutoff of 0.1 to capture more relevant signals”*. We trust the reviewer will appreciate this clarification and rationale.

• *Fig. 3a: there is no scale provided for the heatmap. Are these relative from min to max on a linear scale? Please add to the figure or describe in the legend. Fig. 3c: Please replace the word cloud with some other form of visual representation, or remove altogether and refer to the supplementary table. I am unable to reconcile the differences between the csv file provided and this figure. Based on the table, CHEK2 and several other genes have higher novelty scores than those shown in this figure.*

Thank you very much for your insightful comment. In response to feedback from several reviewers, we have updated our analysis to use a more recent release of the Open Targets Platform (25.03), which includes broader and more up-to-date data. This update altered the list of novel targets, rendering the earlier word cloud based on release 23.06 no longer representative. To ensure the robustness and relevance of our presentation, we replaced Figure 3 with a more granular version of the original Figure 3b, as also suggested in your following comment. We believe the new Figure 3 offers a more general and atemporal overview of novel targets, capturing a three-dimensional perspective: the number of target–disease associations and unique targets identified as novel over the years, stratified by oncological vs. non-oncological categories and by data resource category. This approach better reflects the evolving nature of the data and reduces bias from release-specific snapshots. The new figure is described in “Contributions from high-throughput and clinical resources to biomedical novelty” and “Contributions from expert-curated resources to biomedical novelty” sections in Results. The detailed list of novel targets from the updated release is provided as Supplementary Information to ensure transparency and accessibility. We hope this revised presentation addresses your concerns and enhances both the clarity and scientific rigor of the manuscript. Thank you again for your valuable feedback.

• *Fig. 4: The linear increases in the median differences from peak year to approval year suggests that for most of these evidence sources, most of the peaks occurred in a relatively narrow timeframe. Could you explore this further? E.g. for each source of evidence, the count of peaks by year as you did in Fig. 3b. Probably with and without restrictions to the targets of approved drugs. I believe that this figure is likely to be more informative than the current Fig. 5, so depending on the patterns, you may want to swap that and move Fig. 5 to supplement.*

Thank you for this helpful suggestion. As mentioned in response to your previous comment, we have revised the manuscript to include a more detailed view of the distribution of novelty peaks. Specifically, we have replaced the former Figure 3 with an extended version of Figure 3b, showing the count of peaks by year for each resource category. In addition, a similar figure—depicting the count of peaks by year for each resource—has been added to the Supplementary Information. These new visualisations are now discussed in the Results sections: “Contributions from high-throughput and clinical resources to biomedical novelty” and “Contributions from expert-curated resources to biomedical novelty.” Furthermore, the revised “Retrospective analysis of novel drug targets” section in both Results and Discussion now reflects your observation that “most of the peaks occurred in a relatively narrow timeframe” and explores its possible implications. Finally, as

you suggested, the former Figure 5 has been moved to the Supplementary Information. We hope these updates address your comment and enhance the interpretability and impact of our findings.

• *Data availability: I went to the specified GitHub site and found the code, but not the the underlying data. If code is required to pull the data from an online location, this was not clear. I was expecting to find one or more files that contained the novelty peaks for each gene-trait-year-evidence type, similar to the 2023 example file provided.*

Thank you for raising this point. You are absolutely right. We have now made the data available and accessible via the following Zenodo link: <https://zenodo.org/records/15922783>.

I hope you find this review helpful.

*Warmest regards,
Matthew R. Nelson*

–

Dear Reviewers,

Thank you for your positive feedback on the changes made to my initial manuscript.

Best wishes,

Dr. Maria J. Falaguera

Reviewer #1 (Remarks to the Author):

The authors have adequately answered previous comments.

Reviewer #2 (Remarks to the Author):

All of my initial concerns have been appropriately addressed by the authors.

Reviewer #3 (Remarks to the Author):

The authors have addressed the concerns I raised in the original review and have substantially improved the overall manuscript in addressing the comments of the other reviewers. I have no further comments.